# Microbiome of *Citrullus colocynthis* (L.) Schrad. Reveals a Potential Association with Non-Photosynthetic Cyanobacteria

**DOI:** 10.3390/microorganisms10102083

**Published:** 2022-10-21

**Authors:** Miranda Procter, Biduth Kundu, Naganeeswaran Sudalaimuthuasari, Raja S. AlMaskari, Esam E. Saeed, Khaled M. Hazzouri, Khaled M. A. Amiri

**Affiliations:** 1Department of Biology, College of Science, United Arab Emirates University, Al Ain P.O. Box. 15551, United Arab Emirates; 2Khalifa Center for Genetic Engineering and Biotechnology, United Arab Emirates University, Al Ain P.O. Box. 15551, United Arab Emirates

**Keywords:** colocynth, microbiome profiling, rhizophagy, desert flora, desert soil composition

## Abstract

*Citrullus colocynthis* grows in the sandy desert soil of the Arabian Peninsula with limited access to water, aside from occasional precipitation or dew. Understanding its ability to produce water-filled fruit and nutrient-rich seeds despite the harsh environment, can be useful for agricultural applications. However, information regarding the microbiome of *C. colocynthis* is lacking. We hypothesized that *C. colocynthis* associates with bacteria that aid its survival, like what has been observed in other desert plants. Here, we used 16S rRNA gene data to gain insight into the microbiome of *C. colocynthis* to identify its associated bacteria. In total, 9818 and 6983 OTUs were generated from root, soil, and leaf samples combined. Overall, bulk soils had the highest alpha diversity, followed by rhizosphere and root zone soils. Furthermore, *C. colocynthis* is associated with known plant-growth-promoting bacteria (including Acidobacteria, Bacterioidetes, and Actinobacteria), and interestingly a class of non-photosynthetic Cyanobacteria (Melainabacteria) that is more abundant on the inside and outside of the root surface than control samples, suggesting its involvement in the rhizophagy process. This study will provide a foundation for functional studies to further understand how *C. colocynthis*-microbes interactions help them grow in the desert, paving the path for possible agricultural applications.

## 1. Introduction

Adaptations of desert plants against abiotic stress include succulence, rapid growth when water is available, morphological adaptations (e.g., reduced leaf surface), and many more [1,2]. Additionally, plants interact with microbes to obtain nutrients and minerals from their environment. The microbiome plays a key role in plant health and well-being [3,4]. Desert plants are no exception and the interactions with their microbiome have seen increased interest in recent years [5,6,7], with studies taking place in many deserts around the world [8,9,10,11,12,13]. These studies have ranged from culture- to molecular-based, all showing the incredible microbial diversity that plants associate with to help endure these harsh environments. Most of these studies have identified both epiphytic and endophytic microbes in the category of plant growth promotion, helping plants obtain nutrients and minerals, especially from their surrounding soil. Many studies have also shown that desert-plant-associated microbes are specifically adapted to withstand these harsh environments [6,10].

Microbes in the rhizosphere are vital components of both plant health and nutrition. Previously, the consensus for plants was that they are pure autotrophs, aside from known carnivorous species, such as the Venus Flytrap (*Dionaea muscipula*), pitcher plants (genus *Nepenthes*), etc. [14]. However, research has shown that many plants employ microbivory through rhizophagy [15,16] to access microbe-derived nutrients. During the rhizophagy cycle, microbes from the area surrounding the root tips are taken into plant root cells, and nutrients are extracted from the microbes through an oxidative process. After being cycled around the root cells (cyclosis), the surviving microbes stimulate root hair growth, escaping through the newly developing tips, and the cycle can begin anew [16,17,18]. Rhizophagy is a slow process by which to obtain nutrients such as nitrogen [19]. Nevertheless, it has been suggested that motile bacteria could scavenge nutrients outside their host’s rhizosphere and the root zone, bringing those nutrients back to the plant [16,20]. Through subsequent rhizophagy, this would supply plants with much-needed nutrients that they could not reach otherwise. The benefits of motile microbes and subsequent rhizophagy would be invaluable to desert plants, especially those that produce high amounts of nutrient-rich seeds and water-filled fruits, such as *C. colocynthis*. Other benefits plants may obtain from the rhizophagy cycle include improved stress tolerance through better plant development, increased disease resistance (through pathogen suppression or induced systemic resistance), and enhanced oxidative stress tolerance. In the latter case, increased antioxidant production and upregulation of oxidative stress-related genes are required to ameliorate the negative effects of increased reactive oxygen species (ROS) production in the plant cells, both from the plants and their endophytes [16,20,21].

*Citrullus colocynthis* is a viny desert plant, growing in sandy areas in deserts in parts of Africa, South Asia, Cyprus, Mediterranean Europe, and the Middle East [22,23]. In the United Arab Emirates (UAE), *C. colocynthis* has been reported in the eastern, central, and northern regions of the country, often in sandy areas where there is no visible access to water [24,25]. Most interests in these plants have been related to their medicinal or pharmacological uses and potential as a biofuel feedstock [23,26], with studies on their microbiomes lacking. White et al. [20] suggested that plants such as *C. colocynthis* could potentially rely on motile microbes, and subsequent rhizophagy for nutrient acquisition, allowing these plants to thrive in harsh environments. We hypothesized that *C. colocynthis* associates with microbes to thrive in its environment and aimed to elucidate which bacteria form part of its microbiome, for which information is limited.

In this study, we used 16S rRNA gene data to profile the microbiome of *C. colocynthis* occurring in the UAE and to gain further insights into how this plant survives in this harsh desert environment, most likely with the help of its microbiome.

## 2. Materials and Methods

### 2.1. Sample Collection

Samples were collected in October 2020 from *Citrullus colocynthis* plants in the Nahel area of Al Ain, in the Emirate of Abu Dhabi, United Arab Emirates (24.5195° N, 55.5153° E). The area is characterized by sparse vegetation, sand dunes, and loose sandy planes (Figure 1A), with no rocky areas. During October, the average daytime temperatures in Al Ain range between 34 °C to 40 °C, and chances for precipitation are low (https://weatherspark.com/, accessed on 3 October 2022).

A total of 12 samples were collected from three plants in triplicate, consisting of soil (root zone, bulk soil), leaves, and roots (Figure 1B). For the root zone soil samples, vines were pulled from the soil until a root was discovered, samples were taken from the surrounding soil and transferred to 50 mL tubes (approximating to 50 g soil per tube). Samples were collected within the first 10 to 20 cm of soil, depending on soil cover and subsequent root depth. Thereafter, sections of roots were cut using a sterile scissor and transferred into 50 mL tubes. Bulk soil samples were collected approximately 2 m away from the sampled (and other) plants and transferred into 50 mL tubes. Leaves were randomly cut from different parts of the same plant with a sterile scissor and placed into 50 mL tubes. All samples were placed into individual falcon tubes per replicate and sample type and kept on ice while in the field. After returning from the field on the day of collection, the leaves and roots were surface washed with sterile distilled water, which was collected and transferred to a separate tube. The root surface wash is considered the rhizosphere sample, and the leaf surface wash the phyllosphere sample. All samples were stored at −80 °C in the laboratory until processing.

### 2.2. DNA Isolation and Sequencing

Leaf and root tissue samples were manually ground in liquid nitrogen using a mortar and pestle and kept at −80 °C until extractions were performed. DNA was extracted from the leaf and root tissue using the CTAB method as described by Greco et al. [27]. A skimmed milk protocol adapted from Kashi [28] in combination with the DNeasy PowerSoil kit (Qiagen, Hilden, Germany) was used for the soil samples and the surface wash of the roots and leaves. Briefly, 10 g of soil was mixed with 23 mL cold artificial seawater (pH 3.5) and 0.5% skimmed milk solution, and surface wash samples were mixed with 0.5% skimmed milk solution. Skimmed milk acts as an adsorption competitor, minimizing the amount of DNA adsorbing to soil particles, while artificial seawater provides a slightly saline buffer effect. The mixture was kept overnight on ice packs on a benchtop rocker. A 15 mL tube was filled with the mixture, centrifuged at maximum speed for 5 min, and the supernatant was transferred to a new tube. This step was repeated once more, transferring the supernatant to the same tube. The resulting pellet was used for DNA extraction. The quality of the extracted DNA was confirmed with agarose gel electrophoresis, and quantification was performed using a NanoDrop^TM^ Spectrophotometer (TermoFisher Scientific, Waltham, MA, USA) and Qubit 2.0 fluorometer (TermoFisher Scientific).

DNA samples were amplified using universal V3–V4 primers (16S rRNA gene specific primer 341F: CCTACGGGNGGCWGCAG and 805R: GACTACHVGGGTATCTA ATCC), and sequencing libraries were prepared for the PCR amplicons and sequenced using the Illumina NovaSeq PE250 (100K Paired-end (PE) reads per sample).

### 2.3. 16S rRNA Gene based Microbial Profiling

Raw PE fastq reads were demultiplexed prior to trimming of adapters and primers. DADA2 pipeline was used for quality filtering using the R statistical program [29,30]. Taxonomic assignments were made using Greengenes 16S database reference database [31] at the genus level. Samples were rarefied to a minimum library size of 85783 reads for root, rhizosphere, and soil samples, and 130554 reads for leaf and phyllosphere samples. Alpha and beta diversity were transformed using a logarithmic metric prior to plotting, using Microbiome Analyst [32,33]. We measured Shannon index metric for alpha diversity as well as Operational taxonomic units (OTUs) [34], while Bray–Curtis metric was performed for beta diversity [35]. Statistical differences in alpha and beta diversity were assessed [36] using ANOVA test for alpha diversity and PERMONOVA test for beta diversity with permutation. Heatmaps for class and order level were generated using Microbiome Analyst [32,33].

### 2.4. Soil Composition Analyses

The soil samples were pressed flat and dyed with the aid of aluminium cups (30 mm in diameter) using a manual hydraulic press machine (Retsch, Haan, Germany). Each pellet sample was obtained by mixing approximately 8.0 g of soil uniformly with 2.0 g of Cerox XRF binder (FluXana, Bedburg-Hau, Germany). The pellets were placed in a sample holder cup and were screened using a wavelength dispersive X-ray spectrometer (Rigaku ZSX Primus IV) equipped with a Rh X-ray tube and the instrument is controlled by ZSX Guidance software.

## 3. Results

### 3.1. Species Diversity Indexes

Root, rhizosphere, and soil samples produced 1254130 reads, binned into 9818 OTUs. Samples from leaves and the phyllosphere produced 829218 reads, binned into 6983 OTUs (Appendix A). Diversity indexes grouped samples per sample type, i.e., root, bulk soil, root surface, and root zone soil samples from the three plants grouped together, respectively. These groupings were reflected in the phylum and class level indexes (Appendix A).

In the root and soil samples, alpha diversity at the species level (Figure 2A) showed the most diversity in control (bulk) soil samples, followed by root zone and rhizosphere samples at similar levels. The root tissue samples showed the lowest diversity. The differences between plant and soil samples were reflected in the Principal Coordinates Analysis (PCoA) plot (Figure 2B). The first ordination axis (58.1% of the total variance) separated the species in the root zone and control soil from those in the rhizosphere and root tissues. The second ordination axis (23.8% of the total variance) separated the root zone and two of the rhizosphere samples from the third rhizosphere sample, and the root tissue and control soil samples.

Species diversity was higher in the phyllosphere than in the leaf samples (Figure 2C). The first ordination axis of the PCoA (74.6% of the total variance) separated the leaf and phyllosphere samples, while the second ordination axis (19.4% of the total variance) separated the leaf samples from two of the phyllosphere samples (Figure 2D).

### 3.2. Taxonomic Composition and Abundance

At the phylum level of the roots and soil samples, the most abundant bacterial phyla differed between the different sample types (Figure 3A). The control soil samples had an admixture of different bacteria, with Proteobacteria being the most abundant (37% of total reads), followed by Actinobacteria (25%). Firmicutes were most abundant in the root zone samples (59%), followed by Proteobacteria (23%). The rhizosphere samples differed in the most abundant phyla between the samples. Proteobacteria abundance was relatively consistent (31%), while the abundance levels of Firmicutes and Cyanobacteria differed, with both averaging 24%, respectively, of the total reads for the rhizosphere samples. In the root samples, Cyanobacteria were most abundant (52%), followed by Proteobacteria (37%). The Class-level heatmap (Figure 3B) revealed many taxa not obviously associated with *C. colocynthis* or its immediate surroundings. Compared to the root and soil samples, less diversity was seen in the leaf and phyllosphere samples. Cyanobacteria were the most abundant phylum in both sample types (74% in leaf and 63% in phyllosphere samples), with Proteobacteria (18% in the leaf and 19% in the phyllosphere samples) the second most abundant (Figure 4A). Multiple classes present in the control soil samples were also present in the phyllosphere samples (Figure 4B).

A class of Cyanobacteria (4C0d-2) was present in the rhizosphere and root samples and absent in the root zone and control soil samples. Conversely, another Cyanobacteria class (ML635J-21) was present in the control soil samples and absent in the root zone and root samples. For all three plants, class 4C0d-2 was present either in the rhizosphere or root sample, but not in both. Likewise, class 4Cd0-2 was present in the leaf but absent in phyllosphere samples, and class ML635J-21 was present only in two phyllosphere samples. In the Order-level heatmap (not shown), 4C0d-2 was further classified as MLE1-12. No further subdivisions were identified based on the Greengenes taxonomy database.

One class of Bacteroidetes (Flavobacteria) was present only in the rhizosphere and root samples of plant 2 and showed varied presence between the rhizosphere and root samples of plants 1 and 3. This class was only present in the leaf of one plant, and the phyllosphere of another. Other classes of Bacteroidetes were present mostly in the phyllosphere and control soil samples.

Different classes of Acidobacteria were identified from control and rhizosphere soils, roots, and leaves. Three classes were present only in leaf samples, while others were present in either leaf and control soil samples; leaf, rhizosphere, and root samples; or leaf and root samples only. Only one class (Acidobacteria 5) was shared between leaf, root, rhizosphere, and control soil samples.

Actinobacteria were present in all sample types, but the different classes did not occur in more than one sample type, respectively—i.e., those in the control soil were not present in other sample types, etc.

### 3.3. Soil Composition Analyses

Both root zone and bulk soils were high in calcium (56.3%), and silicon (25.2%). Iron (7.1%), aluminium (3.6%), magnesium (2.6%), and potassium (2.1%) were detected in lower amounts. Very low levels (<1%) of phosphate, sodium, and chlorine were detected (Appendix A). These percentages reflect the average percentage of each element per mass of soil for all soil samples.

## 4. Discussion

Soil conditions in the Arabian Peninsula are known to be very nutrient-limited, high in minerals, such as calcium, magnesium, calcium carbonate, etc., and near neutral pH [5,37]. Nevertheless, microbes and plants can exist in these conditions. Sequencing of 16S rRNA gene from samples of *C. colocynthis* and its surrounding soil has provided more insight into the microbiome of this plant. There was a high diversity in the root and soil samples, while the leaf and phyllosphere samples were less diverse. Proteobacteria, Firmicutes, and Cyanobacteria were the most abundant phyla in the samples, each differing in abundance based on sample type.

Elemental analyses of soil samples indicated that the soils in our sampling area were high in calcium and silicon, and very low in phosphate, sodium, and chlorine, thus the soils are not highly saline. Iron, aluminium, and magnesium were present in lower amounts. High levels of calcium and silicon may be beneficial to plants, as calcium can help ameliorate salt stress effects [38] and silicon can influence the bioavailability of elements such as phosphate and can affect nutrient uptake in plants [39]. The latter may be crucial as the levels of phosphate in the soils were very low (<1%).

Our results revealed that *C. colocynthis* is associated with many known plant-growth-promoting bacteria (PGPB; Figure 3 and Figure 4), including Acidobacteria, Bacteroidetes, and Actinobacteria. Plant growth promoting factors from these bacteria include phosphate solubilization, siderophore and indole-2-acetic acid synthesis, and some bacteria are involved in nitrogen, sulfur, and carbon cycles [40,41,42,43]. These classes were not specific to sample types, with some occurring in both plant and soil samples, but mostly absent from the control soil samples. The occurrence of these PGPB in association with desert plants is known in many desert environments around the world, including the Arabian Peninsula [8,10,12,44]. Association with known PGPB shows that, like most other desert plants, *C. colocynthis* relies on microbial communities to survive in its harsh environment. The presence of siderophore-synthesizing microbes may also be influenced by the low levels of iron detected in the soils. However, there was no significant difference seen between the iron levels in the root zone and bulk soils.

All sample types contained Cyanobacteria. Leaf, root, phyllosphere, and rhizosphere samples showed a high abundance, more so than the control and root zone soil samples. Interestingly, two classes of Cyanobacteria (4C0d-2 and ‘chloroplast’) were prevalent within the rhizosphere and plant tissues, while another Cyanobacteria class (ML635J-2) was present only in the phyllosphere and control soil samples. The presence of ML635J-2 in the control soil and phyllosphere makes sense as the leaves grow in close contact with the soil and are at times completely or partially covered by soil (Figure 1A).

Class 4C0d-2 has been designated as Melainabacteria [45] and class ML635J-2 as Sericytochromatia [46]. Both are non-photosynthetic relatives of the photosynthetic Cyanobacteria, classified as Oxyphotobacteria [46]. Sericytochromatia members have been identified from plant washings, soil, and water in pasture fields [47,48]. On the order level, the Melainabacteria class was classified as MLE1-12, designated as the Obscuribacterales [46]. Melainabacteria consists of four orders, with most identified from human, animal, and insect guts, aside from the order Obscuribacterales. Members of the Obscuribacterales have been isolated from plant washings, soil, bioreactors, aquifers, and lake sediments [45,47,49,50].

Photosynthetic Cyanobacteria (Oxyphotobacteria) provide a plethora of growth-promoting benefits to their host plants and have recently been acknowledged as members of plant microbiomes [48]. However, less is known about the non-photosynthetic lineages, as shown by the extensive review of Lee et al. [51]. Unlike Oxyphotobacteria, Melainabacteria and Serichytochromatia lack the genes for carbon fixation and have been classified as chemoheterotrophs [45,46,49]. Nevertheless, Melainabacteria are capable of nitrogen fixation [52]. A survey of soil Cyanobacteria (photosynthetic and non-photosynthetic) revealed that Oxyphotobacteria were more prevalent in arid and semiarid environments, while Serichytochromatia were more prevalent in hyperarid oligotrophic, and Melainabacteria in acidic or humid, environments [49]. However, the UAE was not included in this survey.

Desert soils are low in nutrients and high in minerals [37] which can be limiting to plant growth. Associating with PGPB is an adaptation by plants to overcome these limiting factors. While the PGPB classes mentioned earlier occurred in either soil or plant samples or both (as has been observed in other desert plants), Melainabacteria (4C0d-2) was present either in the rhizosphere or root samples, but not in both simultaneously. This could be indicative of rhizophagy taking place. Melainabacteria are flagellated and can fix nitrogen [52], possibly providing the plants with much-needed nitrogen in their nitrogen-limited environment, by scavenging from areas other than the root zone. White et al. [20] showed evidence of oxidation and/or degradation of bacteria in the root hairs, cap, and epidermis of *C. colocynthis*. Although the bacteria present in the root cells were not identified, the authors stated that *C. colocynthis* likely partakes in rhizophagy, using bacteria to scavenge nutrients from surrounding soils before taking them up into the root cells and extracting those nutrients through oxidative methods. Partaking in rhizophagy provides essential nutrients and upregulates oxidative stress response in plants, which would help in withstanding the stresses of desert life. These observations may explain the occurrence of Melainabacteria associated with the plant tissues and the rhizosphere (where conditions are more humid) and Sericythochromatia in our control soil samples (where the conditions are arid).

Future work should include isolating PGPB and the non-photosynthetic Cyanobacteria for *in planta* experiments, to further study the interactions between microbe and plant. However, only photosynthetic Cyanobacteria have been cultured to date, with no protocols existing for their non-photosynthetic relatives, which are known only from molecular-based studies. Nevertheless, using what is known from the molecular data on Melainabacteria, and adapting known protocols for the isolation of photosynthetic Cyanobacteria should provide insights into successfully cultivating these microbes. For PGPB, many established protocols are available for use in culturing these microbes.

## 5. Conclusions

This study provides an overview of initial insight into the bacterial microbiome of *C. colocynthis*. Our results revealed that *C. colocynthis* is associated with many known plant growth-promoting bacteria (including Acidobacteria, Bacteroidetes, and Actinobacteria), and interestingly a class of non-photosynthetic Cyanobacteria (Melainabacteria) that is more abundant on the inside and outside of the root surface compared to control samples, which suggest its involvement in the rhizophagy process. Moreover, this study provides a foundation for functional studies to understand further how *C. colocynthis*-microbiome interactions help them thrive in the desert via this process, paving the path for possible applications to agriculture for crop improvement. Future work will include more in-depth molecular analyses using metagenomic and transcriptomic data, in addition to in planta experiments.

## Figures and Tables

**Figure 1 microorganisms-10-02083-f001:**
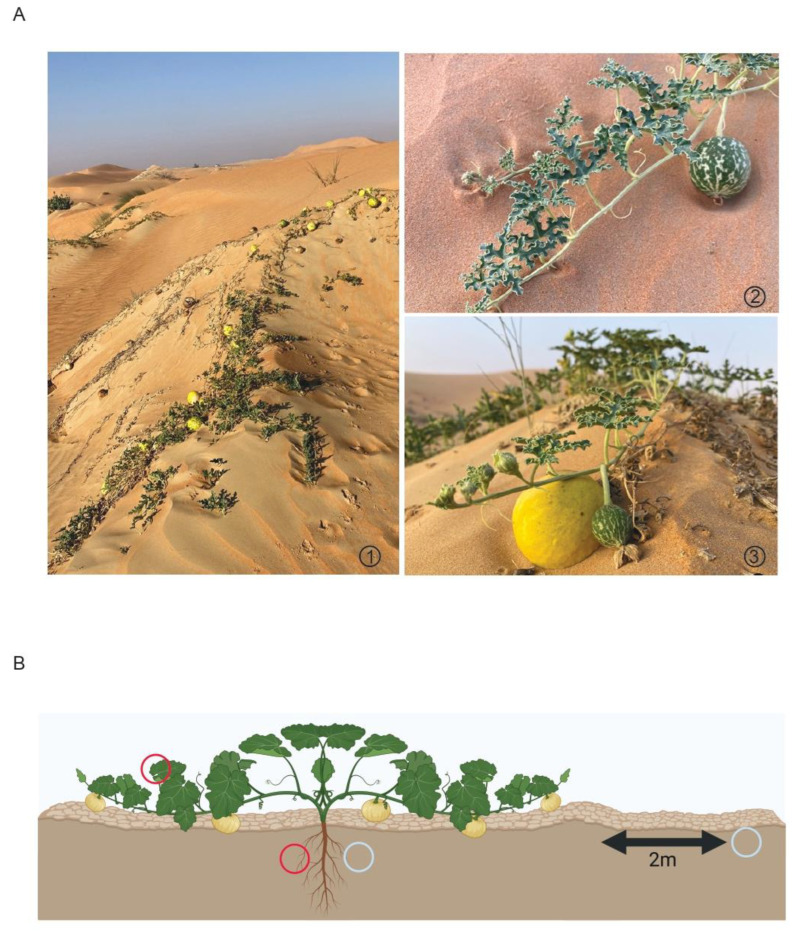
(**A**) *Citrullus colocynthis*. The new season’s growth emerges between dried and dead vines from the previous season ①. Immature fruits are green with irregular white “stripes” ② and newly developing fruit can be seen alongside mature, yellow fruit from the previous season ③. (**B**) Overview of the different samples collected from *C. colocynthis* plants. Red circles represent plant samples, while blue circles represent root zone (closest to roots) and bulk soil (2 m from plant) samples. The image was created with BioRender.com (accessed on 21 September 2022).

**Figure 2 microorganisms-10-02083-f002:**
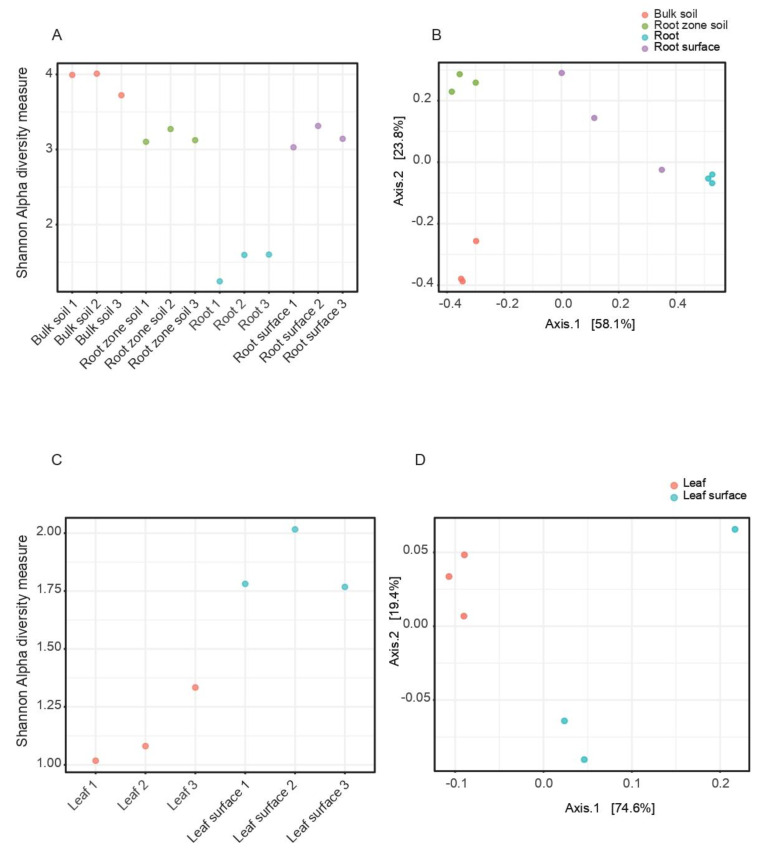
(**A**) Shannon Alpha diversity and (**B**) principal coordinates analysis (PCoA) for beta-diversity (B) at the species level of root and soil samples from three different *C. colocynthis* plants. Alpha diversity metrics (*p*-value: 3.7641 × 10^−7^ (ANOVA) F-value: 131.91); beta diversity metrics ((PERMANOVA) F-value: 13.648; R-squared: 0.83654; *p*-value < 0.001). (**C**) Alpha diversity and (**D**) principal coordinates analysis (PCoA) for beta-diversity (**B**) at the species level of leaf and phyllosphere samples from three different *C. colocynthis* plants. Alpha diversity metrics (*p*-value: 0.0052857; (*t*-test) statistic: −5.652); beta diversity metrics ((PERMANOVA) F-value: 5.5473; R-squared: 0.58103; *p*-value < 0.1).

**Figure 3 microorganisms-10-02083-f003:**
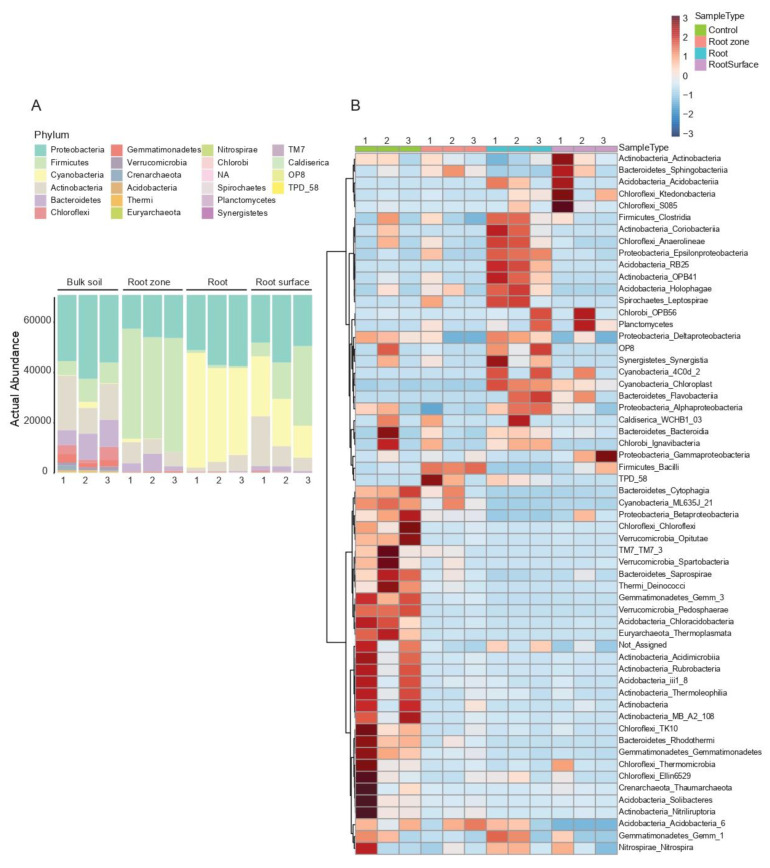
(**A**) Actual abundance at the phylum level of root and soil samples between three different *C. colocynthis* plants. (**B**) Heatmap of taxonomic composition at the Class level of root and soil samples between three different *C. colocynthis* plants. Labels are Phylum_Class, based on Greengenes Taxonomy reference base.

**Figure 4 microorganisms-10-02083-f004:**
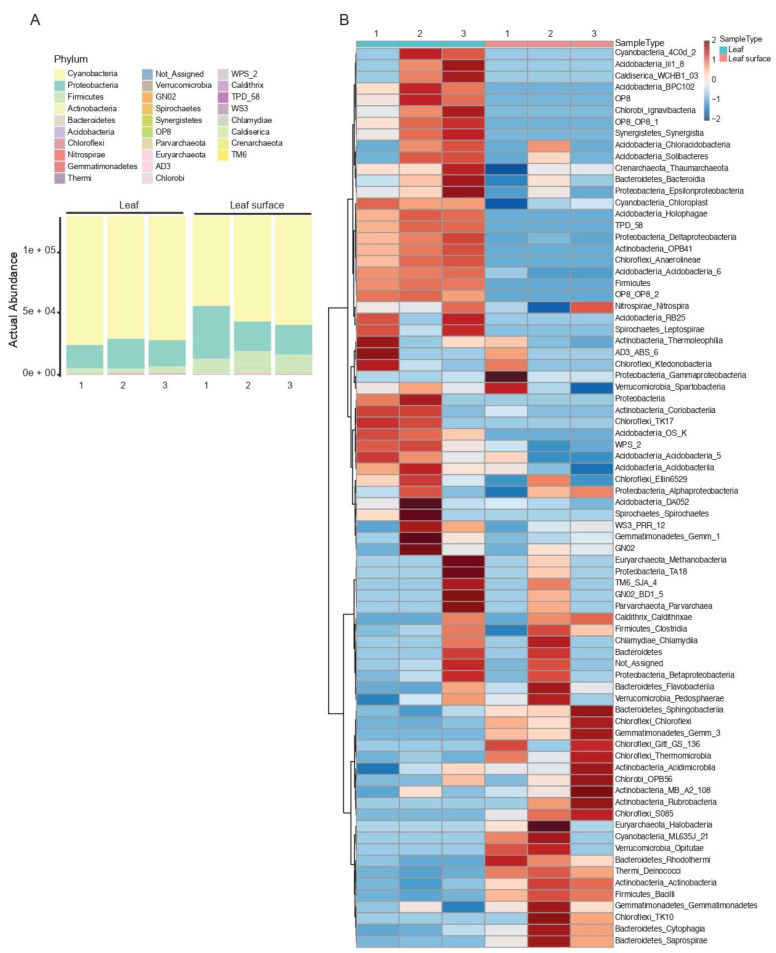
(**A**) Actual abundance at the phylum level of leaf and phyllosphere samples of three individual *C. colocynthis* plants. (**B**) Heatmap of taxonomic composition at the Class level of leaf and phyllosphere samples between three different *C. colocynthis* plants. Labels are Phylum_Class, based on Greengenes Taxonomy reference base.

## Data Availability

The sequencing data generated during this study have been deposited in NCBI-SRA database under the Bioproject ID: PRJNA886871.

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
