# Peer review of "Microbiome of Citrullus colocynthis (L.) Schrad. Reveals a Potential Association with Non-Photosynthetic Cyanobacteria"

_microorganisms, 2022, doi:10.3390/microorganisms10102083_

Round 1

Reviewer 1 Report

A manuscript entitled ‘16S profiling of Citrullus colocynthis (L.) Schrad. (Cucurbitaceae) microbiome reveals a potential association with non-photosynthetic Cyanobacteria’ by Miranda Procter and co-authors for Microorganisms provides information about metagenome studied in soils and Citrullus colocynthis grown on them.

General comments:

I suggest shortening the title of the paper. For example, ‘Microbiome of Citrullus colocynthis (L.) Schrad and feeding soils: a case study in the United Arab Emirates’.

Abstract is too descriptive and does not provide deep insight the main findings.

All keywords are in abstract. Please, use others.

Introduction is too long and descriptive.

Please, add the aim of the research and the scientific hypothesis checked during the research.

Section 2. Please, provide subsection entitled, e.g. ‘Regional setting’ to describe geology, climate, plant cover and soil cover.

Used primers are not specific for chloroplast. I suggest removing the results and discussion connected with data on chloroplast as they are too speculative. Otherwise, present more deep comparative analysis of the results obtained and justify using V3V4 primers to study plant diversity.

Specific comments:

L.11 – 14. The sentence is too long and complicated. Please, simplify.

L.17. Hereinafter. Do you mean 16S rRNA?

L.28 – 30. Please, remove.

L.33 ‘ their environment’. Do you mean ‘solid phase’?

L34 ‘together form the plant’s microbiome’. Please, delete.

L.46. Please, use Latin names.

L.79. Please, provide coordinates of sampling sites.

L.79 – 93. Please, add information about a mean wight of the samples collected.

L.89. How long the samples were transported to a lab?

Section 2.1. Please, add information about soil pretreatment for analysis.

L.107 – 108. I do not understand what the reason to use artificial seawater and skimmed milk. Please, clarify.

L.222. “DAD2”?

Section 3.1 should be earlier.

Figure 2. Please, present differentiation on the phylum and/or class and/or family and/or genus level first. In the text, firstly describe the groups of samples and then compare it.

L.178 ‘‘chloroplast’’. Why? Please, justify.

L.213. I suppose replacing ‘surrounding’ with ‘feeding’.

L.218 ‘PGPB’. What is it?

Section 4. Please, provide illustration with metabolic pathways which are the most abundant in the samples under consideration.

L.269 ‘limiting to plant

Reviewer 2 Report

Author Miranda Procter and colleagues provided a study of associated microbiome to Citrullus colocynthis plantsbased on 16S rDNA microbial profiling.

A net diversity of bacterial communities was revealed associated to Citrullus colocynthis rhizosphere; namely a non-photosynthetic cyanobacteria.

Authors addressed the need of deep study to understand the functional related mechanisms to explain the abundance of such bacterial class in this kind of environment.

The provided paper is well presented with the main required parts. Some modifications are needed to improve the quality of the manuscript:

Line 79: Add indication related to the zone sampling location and characteristics

- Figure 1: Red circles are used to indicate sampling zone. In the manuscript only results about roots and leaves are provided; no fruit sample was analyzed in this study?!

-  Abbreviations must be defined when first cited in the text. Check the manuscript.

- More information about soil characteristics (pH, C and N content, …) could be included to better discuss and conclude about the obtained microbial classes diversity.

Reviewer 3 Report

This manuscript describes an interesting piece of work that shows novel results on Citrullus colocynthis microbiome that could help in the future to elucidate how this plant is able to deal with desert conditions. Nevertheless, the work is quite preliminary and need to address several questions prior to publication.

Summary and introduction sections are correct. Introduction shows properly the state of the art and the motivation for the experimental design of this work, with enough references.

Materials and methods section needs to be improve. My major concern is related with soil characteristics. Authors said “Desert soils are low in nutrients and high in minerals…”, sure that this is true but we should know the characteristics of the soil used in this work (pH, mineral content, organic matter…) to either reproduce your experiments or compare your results with similar works involving desert soils. Did you measure these parameters for the current or previous works?

Line 85: please indicate the depth at which the samples were taken.

Line 89: if samples were just washed with water you cannot be sure that the DNA extracted from roots and leaves corresponded to endophytes since bacteria are nor removed from tissues surface just washing with water. You should have surface disinfected the tissues with hypochlorite and ethanol and checked the lack of bacterial growth after disinfection. Then you can say that the obtained DNA belonged to bacteria from inside roots or leaves. Your samples were contaminated with bacteria from the surface.

Results section also is very vague. Authors just talk about more abundance of bacterial groups, but nothing about percentages of the different groups in each situation. It is interesting to have a quantitative idea of the differences. It could be also useful to represent the same bacterial groups with the same colors in figures 3 and 4. Particularly Cyanobacteria, that appears in yellow in figure 3 and green-blue in figure 4 (in this one yellow is Firmicutes). The referred lack of percentages and differences in colors in the figures make difficult to compare results.

In the discussion section I miss some references and comparison with similar works describing microbiomes of plant in desert regions. It is focused in non-photosynthetic Cyanobacteria and, for example, discussion on PGPB groups is quite poor. Are your results very different from those described for other plants in desert regions? You have to clearly state why you think that non-photosynthetic Cyanobacteria play a key role. Finally, if “Future work will include isolating the Cyanobacteria for in planta experiments” it should be interesting to show at least that you can isolate some of these bacteria and future work is possible.

Round 2

Reviewer 1 Report

The quality of the manuscript was increased. But in the revised version of the manuscript, I did not find the aim of the research and the scientific hypothesis checked during the research. Please, add in the text.

Author Response

We thank the reviewer for their feedback. The following was added previously in the abstract (line 18): “We hypothesized that C. colocynthis associates with bacteria that aid its survival, like what has been observed in other desert plants.”. Following the reviewer’s comment, we have added the following to the introduction (line 80): “We hypothesized that C. colocynthis associates with microbes to thrive in its environment and aimed to elucidate which bacteria form part of its microbiome, for which information is limited.”

Reviewer 3 Report

The authors have answered properly the questions and followed the advices of this reviewer. I think the manuscrito can be published in this improved versión.

Author Response

We appreciate the feedback provided by the reviewer and thank them for their valuable input in improving the manuscript.